# Cervicovaginal levels of human beta defensins during bacterial vaginosis

**Nathalia Mayumi Noda-Nicolau**[1], **Mariana de Castro Silva**[1], **Giovana Fernanda Cosi Bento**[1], **Jeniffer Sena Baptista Ferreira**[1], **Juliano Novak**[1], **Júlia Andrade Pessoa Morales**[1], **Júlia Abbade Tronco**[1], **Aline Nascimento Bolpetti**[1], **Gabriel Vitor Silva Pinto**[1], **Jossimara Polettini**[2], **Camila Marconi**[1,3], **Márcia Guimarães da Silva**[1]*

1 Department of Pathology, Botucatu Medical School, São Paulo State University (UNESP), São Paulo, Brazil, 2 Graduate Program in Biomedical Sciences, Medical School, Universidade Federal da Fronteira Sul (UFFS), Passo Fundo, RS, Brazil, 3 Department of Basic Pathology, Setor de Ciências Biológicas, Universidade Federal do Paraná (UFPR), Curitiba, Brazil

* marcia.guimaraes@unesp.br

## Abstract

### Aims

To compare the cervicovaginal levels of human beta defensin (hBD)-1, 2 and 3 of women according to the status of Nugent-defined bacterial vaginosis (BV).

### Methods

A total of 634 women of reproductive age were included in the study. Participants were equally distributed in two groups: according to the classification of vaginal smears according to Nugent criteria in normal (scores 0 to 3) and BV (scores $\geq$7). Cervicovaginal fluid samples were used for measurements of hBDs1, 2 and 3 levels by enzyme-linked immunosorbent assay (ELISA). Levels of each hBD were compared between the two study groups using Mann-Whitney test, with p-value <0.05 considered as significant. Odds ratio (OR) and 95% confidence interval (95% CI) were calculated for sociodemographic variables and hBD1-3 levels associated with BV a multivariable analysis. Correlation between Nugent score and measured levels of hBDs1-3 were calculated using Spearman's test.

### Results

Cervicovaginal fluids from women with BV showed lower levels of hBD-1 [median 2,400.00 pg/mL (0–27,800.00); p<0.0001], hBD-2 [5,600.00 pg/mL (0–45,800.00); p<0.0001] and hBD-3 [1,600.00 pg/mL (0–81,700.00); p = 0.012] when compared to optimal microbiota [hBD-1: [median 3,400.00 pg/mL (0–35,600.00), hBD-2: 12,300.00 pg/mL (0–95,300.00) and hBD-3: 3,000.00 pg/mL (0–64,300.00), respectively]. Multivariable analysis showed that lower levels of hBD-1 (OR: 2.05; 95% CI: 1.46–2.87), hBD-2 (OR: 1.85; 95% CI: 1.32–2.60) and hBD-3 (OR: 1.90; 95% CI: 1.37–2.64) were independently associated BV. Significant negative correlations were observed between Nugent scores and cervicovaginal levels of hBD-1 (Spearman's rho = -0.2118; p = 0.0001) and hBD-2 (*Spearman's rho = -0.2117; p = 0.0001).

**Data Availability Statement:** The data underlying this study are available in the institutional public repository and can be accessed in http://hdl.handle.net/11449/213676.

**Funding:** This study was supported by the São Paulo Research Foundation (FAPESP; Grant 2012/16800-3, https://fapesp.br), granted to Dr. Márcia Guimarães da Silva of the Department of Pathology. The funders had no role in study design, data collection and analysis, decision to publish, or preparation of the manuscript.

**Competing interests:** The authors have declared that no competing interests exist.

## Conclusions

Bacterial vaginosis is associated with lower cervicovaginal levels of hBDs1-3 in reproductive-aged women.

## Introduction

Human beta defensins (hBDs) are antimicrobial peptides that have an important role in the innate immune response. They are small cationic peptides produced by epithelial and immunity cells. These peptides act by causing electrostatic imbalance leading to pores formation in the microbial membranes and ultimately cell lysis [1]. Still, hBDs have an indirect lysis mechanism by activating microbial intracellular hydrolases that degrade bacterial cell wall or interfere with membrane lipid distribution [2, 3]. Additionally, hBDs interconnect innate and adaptive immune responses of the host through chemotactic action to T-cells, monocytes, dendritic cells and mast cells, and induce the production of proinflammatory cytokines [2, 4, 5]. There are seventeen types of hBDs identified, of which the most studied are hBD-1, -2 and -3 [4–6]. hBD-1 is recognized as the most important antimicrobial peptide of epithelial cells. Its expression is either constitutive or regulated by inflammation in defense against Gram-negative bacteria [7–9]. hBD-2 is effective against Gram-negative and -positive bacteria [10]. Production of hBD-2 is activated by nuclear Factor kappa B (NF-kappa B), signal transducer and activator of transcription 1 (STAT-1) and activator protein 1 (AP-1) transcription factors [11–13]. hBD-3 is also effective against Gram-negative and -positive bacteria [2, 8], and its production is induced by STAT-1 [12] and AP-1 [14, 15] transcription factors, while hBD-3 activation via NF-kappa B is still controversial [12, 14, 15]. In the context of the female genital tract, hBDs contribute for the protection of the vaginal environment against potential pathogens [16] and vaginal epithelial cells are important sources of hBDs [10, 17, 18].

In normal conditions, *Lactobacillus* spp. are the main components of vaginal microbiota. *Lactobacillus* spp. produce lactic acid that leads to an acidic vaginal environment preventing the growth of other bacterial species [19]. In a non-optimal vaginal microbiota *Lactobacillus* spp. are replaced by other bacterial species, mainly facultative and strict anaerobes. Bacterial vaginosis (BV) is the most frequent type of non-optimal vaginal microbiota and it is often diagnosed by microscopic classification of vaginal smears according to Nugent scoring criteria [20]. Prevalence of BV differs across different populations, affecting 30% to 50% of reproductive-aged women worldwide [21–23].

Up to date, few studies have investigated the relation between cervicovaginal levels of hBDs and the components of vaginal microbiota. In the study by Valore et al. [16] vaginal content of antimicrobial polypeptides was found to be strongly related to the capacity of selectively allowing *Lactobacillus* growth and inhibiting other bacterial types. However, the few studies available show conflicting results on cervicovaginal levels of hBDs in relation to BV-status. Thus, many aspects of this relation remain unknown. Given the importance of the maintenance healthy vaginal environment for women´s reproductive health [24, 25], the aim of this study was to compare cervicovaginal levels of hBD-1, -2 and -3 between women with Nugent-BV and those with optimal *Lactobacillus*-dominated microbiota.

## Materials and methods

### Participants and sample collection

This cross-sectional study was performed with 1096 reproductive-aged women who sought Primary Health Care Units in Botucatu, SP, Brazil, for a routine Pap test from September 2012

to January 2013. Eligibility criteria for inclusion in the study were as follows: non-pregnant, non-menopausal, no hysterectomy, no prior report of HIV seroconversion, no use of hormonal or copper intrauterine device (IUD), no vaginal bleeding, no urinary loss, no use of antibiotics or vaginal cream in the preceding 30 days and abstinence from sexual intercourse in the 72 hours preceding the visit. Written informed consent was obtained from all participants prior to enrollment. The study was approved by the Ethics Committee at the Botucatu Medical School (Protocol 02381512.5.1001.5411).

Data on sociodemographic, sexual behavior and gynecological history of participants were obtained by interview. During physical exam, samples were obtained from the mid-vaginal wall using a sterile cotton swab. These samples were smeared on glass slides for microscopic classification according to Nugent's scoring system, in normal (scores 0–3), intermediate (scores 4–6) and BV (scores 7–10) [20], and to detect the presence of *Candida* sp. morphotypes. Samples from the posterior fornix of the vagina were collected to determine infection by *Trichomonas vaginalis* using culture in Diamond´s medium, and endocervical samples were used for detection of *Chlamydia trachomatis* and *Neisseria gonorrhoeae* by PCR. Lastly, 3 mL of sterile saline was used to wash the cervicovaginal region and the liquid was recovered using sterile plastic pipettes. Cervicovaginal fluid samples were centrifuged at 3,000 rpm for 10 min at 4˚C and the supernatant was stored at -80˚C until hBD-1, -2 and -3 assays.

## Detection of *Chlamydia trachomatis*, *Neisseria gonorrhoeae* and *Trichomonas vaginalis*

DNA from cervical samples was extracted using an AmpliLute Liquid Media Extraction Kit (Roche Molecular Systems, Inc.) according to manufacturer´s instructions. *C. trachomatis* and *N. gonorrhoeae* were detected by polymerase chain reaction, as previously described [26, 27]. Infection by *T. vaginalis* was investigated by culture of vaginal posterior fornix samples in Modified Diamond's medium at 37˚C in 5% $CO_2$. Using a clean glass slide without cover and a sterile pipette, fresh wet-mount microscope slides were prepared with aliquots from the culture and examined microscopically under x40 objective. Presence of the motile protozoan was checked daily up to 3 days by the same observer if *T. vaginalis* motile protozoan was not visualized during this period, the specimen was considered negative.

## Quantification of cervicovaginal levels of hBDs

Cervicovaginal fluid samples were evaluated by ELISA using PeproTech specific kits (Rocky Hill, NJ, USA) to measure hBD-1 (cat#900-M202), hBD-2 (cat#900-M172) and hBD-3 (cat#900-M210) levels, following the manufacturer's instructions. All samples were tested in duplicate. Samples in which hBD levels were estimated to be below the sensitivity of the assay were set as zero, and those with concentrations at levels above standard curve were diluted and re-assayed. The assays were analyzed in an automatic microplate reader (Epoch-BioTek, Winooski, VT, USA), at a wavelength of 492 nm. Intra- and inter-assay coefficients of variation were <5% in this study. Internal laboratory quality-controls assurance program including external and internal standards was followed for all analysis. Minimum detectable levels were 60.00 pg/mL, 130.00 pg/mL, and 4.00 pg/mL for hBD-1, hBD-2 and hBD-3, respectively.

## Participant selection and constitution of the study groups

Of the 1096 participants initially included, 37 (3.4%) were excluded by presenting *C. trachomatis* and 10 (0.9%) for *T. vaginalis*. None of the participants tested positive for *N. gonorrhoeae*. Moreover, 112 cases were excluded as microscopic analysis of vaginal microbiota showed other alteration besides BV, as follows: 39 (3.6%) presence of *Candida* spp. morphotypes, 5

(0.5%) concomitant BV and *Candida* spp., 62 (5.6%) intermediate microbiota, 6 (0.6%) poor sample quality. Among the 937 remaining participants, 317 were diagnosed with BV and were assigned the study group ´BV´. In order to constitute the control group ´optimal microbiota´, same number of participants (n = 317) were randomly selected amongst the 620 participants that had normal vaginal microbiota.

Results of continuous variables were compared between the two study groups using Mann-Whitney non-parametric test, while those from categorical variables were compared by Chi-square test. P-value inferior to 0.05 considered statistically significant. A multivariable logistic regression analysis was performed using a forward stepwise model selection process (variables retained at P-value ≤0.15) to test the variables independently associated with BV from socio-demographic and hBD1-3 data available. For this analysis we considered as 'low' hBDs levels those inferior to the median (2,795.00 pg/mL, 8,335.00 pg/mL and 2,054.00 pg/mL for hBD-1, 2 and 3, respectively). The correlation between Nugent score and measured levels of hBDs1-3 were assessed statistically by calculating Spearman correlation coefficients (Spearman's rho), for those participants from BV group. Statistical analyses were performed using Stata/SE 15.1 (StataCorp, College Station, TX).

## Results

Participant characteristics and clinical history are presented in Table 1. The variables age, marital status, ethnicity, educational status, sexual behavior, vaginal intercourse in the last week and phase of menstrual cycle did not differ between the two study groups. However, the proportion of women that used hormonal contraceptive was significantly higher in optimal microbiota (p = 0.04). Additionally, smoking and higher vaginal pH were more frequent in BV (p = 0.03 and p<0.0001, respectively).

The results of cervicovaginal hBDs quantification showed that approximately 10% of the samples presented levels below the limit of detection for hBD-1 and 2. Levels of hBD-3 were undetected in nearly 20% of the sample from both study groups. Fig 1 shows the comparisons of hBD levels between BV and optimal microbiota. Significantly lower cervicovaginal levels of all the three hBDs were observed in BV in relation to optimal microbiota, respectively: hBD-1: 2,400.00 pg/mL (0–27,800.00) *vs*. 3,400.00 pg/mL (0–35,600.00), p<0.0001; hBD-2: 5,600.00 pg/mL (0–45,800.00) *vs*. 12,300.00 pg/mL (0–95,300.00), p<0.0001; hBD-3: 1,600.00 pg/mL (0–81,700.00) *vs*. 3,000.00 pg/mL (0–64,300.00), p = 0.012.

Multivariable analysis showed that sex partners in the previous year (OR: 1.47 95% CI: 1.06–2.05), and lower levels of hBD-1 (OR: 2.05; 95% CI: 1.46–2.87), hBD-2 (OR: 1.85; 95% CI: 1.32–2.60) and hBD-3 (OR: 1.90; 95% CI: 1.37–2.64) were overrepresented among BV-positive women. Women who reported using hormonal contraceptive were less likely to have BV (OR: 0.55; 95% CI 0.39–0.78) (Table 2).

There were significant negative correlations between Nugent scores and cervicovaginal levels of hBD-1 (Spearman's rho = -0.2118; p = 0.0001) and hBD-2 (*Spearman's rho = -0.2117; p = 0.0001), but not for hBD-3 (Spearman's rho = -0.0600; p = 0.3181).

## Discussion

The association between hBDs and BV in reproductive-aged women has been poorly studied to date, and most of them are based on a very limited sample size [18, 28]. Our study included a total of 634 measured samples and showed that BV has lower levels of hBD-1, -2 and -3.

Regarding sociodemographic, behavior and clinical characteristics, it is documented the importance of several factors as smoking, hormonal status, menstrual cycle period and sexual behavior with increased risk for BV [29–33]. In the current study we observed a significant

**Table 1. Sociodemographic, behavior and clinical characteristics of study population, according to the status of the vaginal microbiota.**

| Variables | Optimal microbiota (n = 317) | Bacterial vaginosis (n = 317) | p-value |
|---|---|---|---|
| **Age [a]** | 35 (16–53) | 35 (14–54) | 0.91 |
| **Marital status[b]** | | | |
| Single | 238 (75.1) | 220 (69.4) | 0.13 |
| In a steady relationship | 79 (24.9) | 97 (30.6) | |
| **Ethnicity (self-reported) [b]** | | | |
| White | 196 (61.8) | 193 (60.9) | 0.87 |
| Non-white | 121 (38.2) | 124 (39.1) | |
| **Years at school[a]** | 11 (0–18) | 10 (0–17) | 0.28 |
| **Number of sex partners, (last 12 months) [b]** | | | |
| 0 or 1 | 301 (94.5) | 294 (92.8) | 0.32 |
| 2 or more | 16 (5.5) | 23 (7.2) | |
| **Consistent condom use [b]** | | | |
| Yes | 48 (15.2) | 49 (15.4) | 1.00 |
| No | 269 (84.8) | 268 (84.6) | |
| **Number of vaginal intercourse/week [b]** | | | |
| 0 | 28 (8.9) | 29 (9.1) | 0.81 |
| 1–2 | 152 (47.9) | 159 (50.2) | |
| 3+ | 137 (43.2) | 129 (40.7) | |
| **Hormonal contraceptive current use [b]** | | | |
| Yes | 131 (41.3) | 105 (33.1) | 0.04 |
| No | 186 (58.7) | 212 (66.9) | |
| **Menstrual cycle phases [b]** | | | |
| Follicular | 84(26.5) | 97 (30.6) | 0.47 |
| Ovulation | 16 (5.0) | 13 (4.1) | |
| Luteal | 217 (68.5) | 207 (65.3) | |
| **Smoking habit[b]** | | | |
| Yes | 43 (13.6) | 64 (20.2) | 0.03 |
| No | 274 (86.4) | 253 (79.8) | |
| **Vaginal pH[a]** | 4.4 (4.0–5.0) | 4.7 (4.0–7.0) | <0.0001 |

[a] median (minimum—maximum), non-parametric Mann-Whitney U test.

[b] n (%), Chi-square test.

difference regarding smoking habit and hormonal contraceptive use between BV-positive and -negative women, but when we performed a multivariable analysis only the hormonal contraceptive use remained significant. The latter analysis also showed other factors significantly associated with BV as having 3 or more sex partners in the previous year and lower hBD-1, 2 and 3 levels. Interestingly, the menstrual cycle phases were not significant different between the groups. Corroborating our findings, Valore et al. [17] demonstrated significant lower levels of hBD-1 and -2 in women presenting BV compared to healthy women. Recently, Fichorova et al. [34] reported decrease of hBD-2 in Nugent scores of 4 to 6 and 7 to 10, although cervical immunity detectable within 3 months prior to cervicovaginal infection or dysbiosis showed higher hBD-2 levels. In contrast, Fan et al. [28] reported an increase of vaginal hBD-2 in the presence of BV in non-pregnant women. During pregnancy, Mitchell et al. [35] detected lower vaginal levels of hBD-3 in BV, but no difference was observed regarding hBD-2 levels. On the other hand, a lower level of hBD-2 was reported by Kotani et al. [36] in the first trimester of pregnancy in BV-positive women. Experimental data inducing cervical hBD-3 gene expression

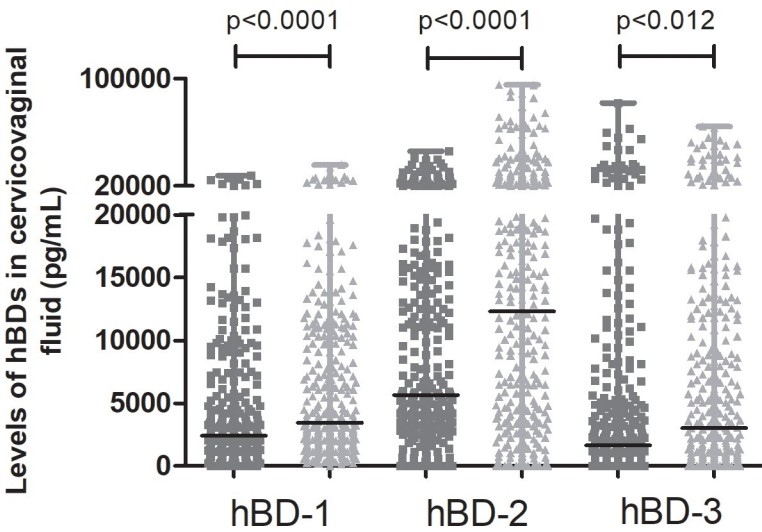

**Fig 1. Levels of human beta defensin (hBD)-1, 2 and 3 in cervicovaginal fluid samples of women with bacterial vaginosis (n = 317) compared to those with optimal microbiota (n = 317).** Horizontal bars represent median values. Mann-Whitney U test, $p < 0.05$.

in a mouse model of ascending infection-related preterm birth demonstrated that hBD-3 reduces microbial ascension into the pregnant uterine cavity, reducing the frequency of premature deliveries [37]. These findings reinforce that hBDs may be potential candidates for augmenting cervical innate immunity, to prevent ascending infection and to reduce susceptibility to sexually transmitted infections, which is supported by the observations that antimicrobial activity is increased in samples with higher hBD-2 concentrations [38]. Therefore, BV and other vaginal dysbiosis seem to be related to lower hBD production, which is reinforced by the current observations.

According to the literature, it is well established that the presence of *Lactobacillus* spp. in healthy vaginal microbiota has an important role in competitive exclusion of pathogenic bacteria, competition for nutrients, production of antimicrobial substances, and n of the immune system [24, 39]. In this context, Jiang et al. [40] showed a significant correlation between hBD-2 and hBD-3 and DNA levels of *L. jensenii*, and hBD-2 and DNA levels of *L. crispatus* in cervicovaginal lavage samples from healthy women. Accordingly, Kotani et al. [18] reported the Nugent score for *Lactobacillus* morphotype point was well correlated with hBD-2. Additionally, studies using intestinal epithelial cell cultures demonstrate that *Lactobacillus* spp. are able to up-regulate hBD-2 production [41, 42] by induction of proinflammatory pathways, such as NF-kappa B and AP-1, as well as MAPKs [41]. Therefore, since our results showed decreased levels of hBD-1, -2 and -3 in the presence of BV, a scenario characterized by the absence of *Lactobacillus* species, we hypothesized that the optimal microbiota had an important role in human beta defensins production. In this sense, a recent study showed that *Lactobacillus* surface layer proteins (SLP) stimulate the expression of antimicrobial peptides, specifically, SLP of *Lactobacillus helveticus* SBT2171 promotes hBD-2 expression by activating c-Jun N-terminal kinase (JNK) signaling via Toll-like receptor (TLR)2 in Caco-2 human colonic epithelial cells [43]. Thus, considering the high diversity of bacterial components of the vaginal microbiota, especially during BV, we acknowledge that our study is limited as Nugent scoring does not allow us to determine which bacterial species are present and what their individual association with hBD1-3.

**Table 2. Odds ratio (OR) and 95% confidence intervals (95% CI) for the association between sociodemographic, and low human beta-defensis (hBDs) 1 to 3 cervicovaginal levels and Nugent-bacterial vaginosis.**

| | | Multivariable analysis | | |
| --- | --- | --- | --- | --- |
| | | OR | 95% CI | P-value |
| **Age** | | -- | -- | -- |
| | <35 years | | | |
| | ≥35 years | | | |
| **Marital status** | | -- | -- | -- |
| | Single | | | |
| | In a steady relationship | | | |
| **Ethnicity** | | -- | -- | -- |
| | White | | | |
| | Non-white | | | |
| **Education level** | | -- | -- | -- |
| | Uncompleted high school | | | |
| | Completed high school or above | | | |
| **Number of sex partner** (last 12 months) | | | | 0.022 |
| | 0 to 2 | 1.47 | 1.06–2.05 | |
| | 3 or more | | | |
| **Consistent condom use** | | -- | -- | -- |
| | Yes | | | |
| | No | | | |
| **Frequency of sexual intercourse (per week)** | | -- | -- | -- |
| | 0 to 2 | | | |
| | 3 or more | | | |
| **Hormonal contraceptive use (current)** | | | | 0.001 |
| | Yes | 0.55 | 0.39–0.78 | |
| | No | 1.00 | -- | |
| **Phase of menstrual cycle** | | -- | -- | -- |
| | Follicular | | | |
| | Luteal/ovulatory/continous hormonal contraceptive | | | |
| **Smoking habit** | | -- | -- | -- |
| | Yes | | | |
| | No | | | |
| **Low hBD-1 level*** | | | | <0.0001 |
| | Yes | 2.05 | 1.46–2.87 | |
| | No | 1.00 | -- | |
| **Low hBD-2 level*** | | | | <0.0001 |
| | Yes | 1.85 | 1.32–2.60 | |
| | No | 1.00 | -- | |
| **Low hBD-3 level*** | | | | <0.0001 |
| | Yes | 1.90 | 1.37–2.64 | |
| | No | 1.00 | -- | |

*Low levels of HBD were set for values inferior to the median.

In addition, once BV is characterized by a complex and heterogeneous bacterial community, it may be that bacterial proteases could degrade hBDs [44], which provides an environment that favors the degradation of natural antimicrobial peptides by the diverse proteases

produced by different bacterial species [44]. This environment could explain why, even in the presence of proinflammatory cytokines, such as IL-1β [32], which are able to stimulate hBDs production [45], levels of these antimicrobial peptides in women with BV are lower than in optimal microbiota.

Thus, we may hypothesize that the lower levels of hBD-1, -2 and -3 in the presence of BV could be due to the production of proteases by the highly diverse bacterial community of BV and because of the lack of *Lactobacillus* species in the vaginal microbiota which could alter the secretion of defensins. However, the mechanisms by which vaginal *Lactobacillus* spp. and epithelial cells interact to enhance the immune response by increasing hBDs production should be further investigated. Moreover, a multivariate analysis has been carried out to explore the nature of the associations observed in the univariate analysis and the levels of hBD 1–3 remain significantly associated with BV as do the number of sexual partners and use of hormonal contraception. Then, we suggest additional future analyses considering this factors, which might provide mechanistic insights into the associations that we described.

## Conclusions

Since women with BV had lower cervicovaginal levels of hBD-1, -2 and -3, compared to women with optimal microbiota, we suggest that the presence of *Lactobacillus* spp. in the vaginal microbiota plays an important role in the production of hBDs in women of reproductive age.

## Acknowledgments

We would like to thank all the patients of this study.

## Author Contributions

**Formal analysis:** Nathalia Mayumi Noda-Nicolau, Camila Marconi.

**Funding acquisition:** Márcia Guimarães da Silva.

**Investigation:** Nathalia Mayumi Noda-Nicolau, Mariana de Castro Silva, Giovana Fernanda Cosi Bento, Jeniffer Sena Baptista Ferreira, Júlia Abbade Tronco, Aline Nascimento Bolpetti, Gabriel Vitor Silva Pinto.

**Methodology:** Nathalia Mayumi Noda-Nicolau, Juliano Novak, Júlia Andrade Pessoa Morales, Aline Nascimento Bolpetti, Gabriel Vitor Silva Pinto.

**Project administration:** Márcia Guimarães da Silva.

**Supervision:** Márcia Guimarães da Silva.

**Writing – original draft:** Nathalia Mayumi Noda-Nicolau, Camila Marconi.

**Writing – review & editing:** Jossimara Polettini, Márcia Guimarães da Silva.

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
