## [Decision Letter · Decision Letter 0]

26 Mar 2021

PONE-D-21-03534

Cervicovaginal levels of human beta defensins during bacterial vaginosis

PLOS ONE

Dear Dr. Guimaraes da Silva,

Thank you for submitting your manuscript to PLOS ONE. After careful consideration, we feel that it has merit but does not fully meet PLOS ONE’s publication criteria as it currently stands. Therefore, we invite you to submit a revised version of the manuscript that addresses the points raised during the review process.

We ask you in particular to perform additional statistical analysis to account for known confounders, to review more rigorously the exisiting literature and to discuss it,  and if possible to include addititional data on pro-inflammatory cytokines.

The method section should provide more details on the optimization and verification of the HBD assays, in particular for what concerns the lower detection levels and reproducibility. 

We look forward to receiving your revised manuscript.

Kind regards,

Tania Crucitti

Academic Editor

PLOS ONE

Journal Requirements:

Reviewers' comments:

Reviewer's Responses to Questions

**Comments to the Author**

1. Is the manuscript technically sound, and do the data support the conclusions?

Reviewer #1: Partly

Reviewer #2: Partly

2. Has the statistical analysis been performed appropriately and rigorously? 

Reviewer #1: No

Reviewer #2: No

3. Have the authors made all data underlying the findings in their manuscript fully available?

Reviewer #1: Yes

Reviewer #2: Yes

4. Is the manuscript presented in an intelligible fashion and written in standard English?

Reviewer #1: Yes

Reviewer #2: Yes

5. Review Comments to the Author

Reviewer #1: This study builds on our understanding of changes in immune mediator concentrations associated with bacterial vaginosis (BV), which is a key risk factor for sexually transmitted infection (STI) acquisition, including HIV, and adverse reproductive outcomes. A strength of this work is that a large sample size of 634 women was included. hBDs-1, -2 and -3 were measured in duplicate, another strength, in all samples. Although women with common STIs, intermediate microbiota or candidiasis were excluded, the analysis presented would benefit from multivariate approaches to identify and address other possible confounders and effect modifiers. A clear description of the quality control analysis would also be useful.

1. Although the manuscript is generally clear, concise and flows well, there are (mostly minor) grammatical errors throughout and I would recommend copy editing.

2. Page 6: It is reported that women with C. trachomatis, T. vaginalis, candidiasis or intermediate microbiota were excluded. Although N. gonorrhoeae was assessed, it is not reported whether any women had this infection or if women with this infection were also excluded.

3. Page 6, line 134: “Minimum detectable levels were 0.06 pg/mL, 0.13 pg/mL, and 0.004 pg/mL for hBD-1, hBD-2 and hBD-3, respectively”. This seems very low, particularly for hBD-3. Standard curves shown on the manufacturer’s website show that hBD-3 seems to plateau before 10pg/ml. Can the authors confirm the accuracy of measurements in the <10 pg/ml range?

4. Page 6, line 138: Please clarify the following: “37 (3.4%) were excluded by presenting

C. trachomatis or T. vaginalis (n=10, 0.9%)”. Does the “37” refer to C. trachomatis and the “10” refer to T. vaginalis? This is not written clearly.

5. Were any women using the copper IUD – only hormonal methods and condoms are mentioned?

6. The authors should adjust p-values for multiple comparisons where applicable.

7. Since the hBDs were measured in duplicate, it would be useful to present some data showing the reproducibility of the measurements – e.g. CVs

8. Were hBDs associated with any demographic/socio-behavioural factors assessed in this study? The authors should assess this and evaluate the relationship between hBDs and BV, adjusting for possible confounders and effect modifiers e.g. hormonal contraceptive use and smoking differ between comparison groups.

9. Page 9-11: The following conclusions should be reworded:

“Thus, the production of hBDs is related to the presence of Lactobacillus spp. in the vaginal microbiota.”

“but is mainly due to the lack of Lactobacillus species in the vaginal microbiota which could modulate the secretion of defensins”

“we suggest that the presence of Lactobacillus spp. in the vaginal microbiota plays an important role in the regulation of human beta defensins production in women of reproductive age.”

It is not possible, based on the data presented in this study (i.e. Nugent score) to conclude that hBD production is related to Lactobacillus spp. specifically. The relationship may have nothing to do with the lactobacilli, but may rather be due to downregulation or degradation of hBD production in the presence of non-optimal bacteria (which is discussed on page 11). It would also be useful to compare the score for the Lactobacillus morphotype to hBD concentrations – a sub-analysis could also be conducted to evaluate the relationship between Lactobacillus score and hBD concentrations among BV negative women only (i.e. when Gardnerella and Mobiluncus morphotypes are absent or rare, is hBD associated with Lactobacillus morphotype score?).

10. Page 10, line 217: Please clarify the following statement regarding “activation of the immune system” by Lactobacillus spp. since lactobacilli are also considered immunoregulatory.

11. It would be preferable to refer to “optimal” microbiota, instead of “normal” microbiota. See McKinnon et al. AIDS Research and Human Retroviruses. 2019 Mar 1;35(3):219-28.

Reviewer #2: The authors measured the levels of HBD1, 2 and 3 in CVL from 634 women by BV status defined by Nugent score. They show that the levels of all three defensins to be significantly lower in 317 women with BV (Nugent score of >7) than in 317 randomly selected women without BV (Nugent score 0-3). BV is associated with poor health outcomes and increased susceptibility to HIV which has been associated with pro-inflammatory responses and there is great interest in gaining a better understanding of the etiology.

It is argued that via their direct antimicrobial action, the beta defensins contribute to the protection of the vaginal microenvironment. There is circumstantial evidence that the innate immune response to resident Lactobacilli which includes anti-microbial peptides produced by the vaginal epithelium, contributes to protection against bacterial colonisation and subsequent dysbiosis associated with BV. The authors cite evidence that vaginal antimicrobial polypeptides can selectively allow for the growth of (resident) Lactobacilli, which is suggestive of a mechanism but the data is derived from a small number of women.

Existing data on the association with BV and levels of HBDs is limited and somewhat contradictory with some studies showing higher levels of HBD and some showing lower levels. HBDs recruit innate and adaptive immune cells and associated proinflammatory cytokines such as IL6 and IL8 have also been measured in some studies- again with conflicting results. In a review from 2014 which is not cited by the authors, Mitchell and Marazzo discuss the immune response associated with BV and any evidence for the modulating impact of Lactobacilli in the context of inflammatory markers (associated with increased susceptibility HIV). They review data on the associations of the proinflammatory cytokines ILbeta and IL8 with BV and also discuss associations with antimicrobial peptides. There is no consistent finding and they conclude that the considerable variability in findings is probably due to the fact that most studies are small and underpowered and constrained by their cross-sectional design which does not account for the acknowledged (almost daily) fluctuations in vaginal microbiota (Gajer 2012). Longitudinal treatment studies are considered more robust and provide more consistent data with 5/6 showing reductions in levels of IL1beta and 3/6 reductions in IL8 after treatment for BV. Data on HBD was derived predominantly from pregnant women (Mitchell et al 2013) and showed that all 9 bacterial species associated with BV were associated with lower levels of HBD3 – but this was not true for HBD2. In a more direct approach, genetic analysis has also shed some light on factors associated with susceptibility to BV with polymorphisms in TLR IL6 and IL8 associated with increased susceptibility to BV arguing a role a robust innate immune response in protection. Whilst data on cytokines and BV has also been conflicting, this reviewer thinks that the paper would be strengthened by inclusion of data on the levels of proinflammatory cytokines such as IL1b, IL6 and IL8 as this would provide insights into the immunological consequences of the lower observed levels of HBD1, 2, 3 and potentially provide some mechanistic insights which have been suggested.

The etiology of BV is undoubtedly complex and multifactorial –without even considering the diversity in bacterial species which have been associated with both health and disease across different studies which complicates the interpretation of associations seen. The authors have not addressed this complexity and the gaps in their review of the literature reflects this. In their review Martin and Marazzo (JID 2014) discuss the importance of accounting for smoking, hormonal status and sexual behaviour in analyses as these are all known to play a role (Mitchell and Murphy 2014). In particular, estrodiol has been shown to be reduce the production of HBD2 and SLP and mRNA expression of TLR by vaginal epithelial cells in vitro (Wira et al 2015). In their comprehensive paper of 2019, Fichorova et al also describe an important role for hormonal contraception in susceptibility to dysbiosis in the context of innate immune responses. Inflammatory responses are also known to fluctuate throughout the menstrual cycle with lower levels from day 7-10. These authors show that levels of smoking and hormonal contraceptive use differ significantly by BV status and this could confound their analysis. The authors state that they cannot access any more information about hormonal contraceptive use, but they should at least discuss the limitations of their study in the context of available literature. At the very least they should run a multivariate analysis to accounts for smoking and hormonal status to see whether the associations with BV and HBD remain significant.

It is acknowledged that BV is characterised by the general absence of “healthy Lactobacilli “ and it is debated whether the lactobacilli exert a positively protective effect. It has been shown for example that L crispatus and L jensenii are able to directly induce the production of HBD2 and thereby inhibit the growth of some gram negative bacteria. The reviewer understands that it is probably beyond the scope of the authors to undertake a comprehensive characterisation of the bacterial composition within their cohort, but this should be addressed within the discussion.

In conclusion, this cross sectional study provides some interesting suggestive evidence that anti-microbial defensins might be associated with BV but limitations in the design mean that it provides limited additional extra value to the existing literature in its current form . The paper would be benefit from additional analysis to account for known confounders,a more thorough review of the literature discussing limitations, and if at all possible inclusion of additional data on levels of the pro-inflammatory cytokines which have been shown to be associated with BV.

6. PLOS authors have the option to publish the peer review history of their article (what does this mean?). If published, this will include your full peer review and any attached files.

Reviewer #1: No

Reviewer #2: No

---

## [Author Response · Author response to Decision Letter 0]

28 Jul 2021

RESPONSE TO REVIEWERS

Comments to the Author

1. Is the manuscript technically sound, and do the data support the conclusions?

Reviewer #1: Partly

Reviewer #2: Partly

2. Has the statistical analysis been performed appropriately and rigorously?

Reviewer #1: No

Reviewer #2: No

3. Have the authors made all data underlying the findings in their manuscript fully available?

Reviewer #1: Yes

Reviewer #2: Yes

4. Is the manuscript presented in an intelligible fashion and written in standard English?

Reviewer #1: Yes

Reviewer #2: Yes

5. Review Comments to the Author

Reviewer #1: This study builds on our understanding of changes in immune mediator concentrations associated with bacterial vaginosis (BV), which is a key risk factor for sexually transmitted infection (STI) acquisition, including HIV, and adverse reproductive outcomes. A strength of this work is that a large sample size of 634 women was included. hBDs-1, -2 and -3 were measured in duplicate, another strength, in all samples. Although women with common STIs, intermediate microbiota or candidiasis were excluded, the analysis presented would benefit from multivariate approaches to identify and address other possible confounders and effect modifiers. A clear description of the quality control analysis would also be useful.

Answer: We appreciate the careful review and enriching comments made by this reviewer. All points were discussed by the authors and answers and comments are provided bellow. 

1. Although the manuscript is generally clear, concise and flows well, there are (mostly minor) grammatical errors throughout and I would recommend copy editing.

Answer: In fact we have detected some writing errors and have rewritten according. 

2. Page 6: It is reported that women with C. trachomatis, T. vaginalis, candidiasis or intermediate microbiota were excluded. Although N. gonorrhoeae was assessed, it is not reported whether any women had this infection or if women with this infection were also excluded. 

Answer: We thank this reviewer for pointing that out. None of the participants tested positive for Neisseria gonorrhoeae, as now stated in page 6 (results section).

3. Page 6, line 134: “Minimum detectable levels were 0.06 pg/mL, 0.13 pg/mL, and 0.004 pg/mL for hBD-1, hBD-2 and hBD-3, respectively”. This seems very low, particularly for hBD-3. Standard curves shown on the manufacturer’s website show that hBD-3 seems to plateau before 10pg/ml. Can the authors confirm the accuracy of measurements in the <10 pg/ml range? 

Answer: We thank the reviewer for this pointing this out. We did not realize that the measure unity was in ng/mL instead of pg/mL. Therefore, we corrected the data multiplying the measured values by a factor of 1000 so it is now correctly expressed in pg/mL. Minimum values are very close to manufacture’s standard curve interval. We regret the submitted manuscript had this mistake and thank the reviewer for the careful analysis. All values were corrected for the newly version of the submitted paper. 

4. Page 6, line 138: Please clarify the following: “37 (3.4%) were excluded by presenting

C. trachomatis or T. vaginalis (n=10, 0.9%)”. Does the “37” refer to C. trachomatis and the “10” refer to T. vaginalis? This is not written clearly. 

Answer:We thank this reviewer for pointing this issue. In fact, the text was not sufficiently accurate. We corrected the text for “37 (3.4%) were excluded by testing positive for presenting C. trachomatis and 10 (0.9%) for T. vaginalis” (page 06) at the results section of the manuscript.

5. Were any women using the copper IUD – only hormonal methods and condoms are mentioned? 

Answer We appreciate this reviewer mentioning the possibility of having IUD-users in our study population. We did not considered for study enrollment those women who reported IUDs of any type (cooper or hormonal), because their known association with BV. We understand that this should be clearly stated in the text. Please see the text added to page 4 (line 106) for addressing this topic. 

6. The authors should adjust p-values for multiple comparisons where applicable.

Answer: We do agree with this reviewer that multiple comparisons is relevant and should be included in the analysis. Thus, we performed a multivariable logistic regression analysis using a forward stepwise model selection process (variables retained at P-value ≤0.15) to identify sociodemographic and hBDs variables independently associated with BV from sociodemographic and hBD1-3 data available. This analysis is presented in Results section pages 09-10.

7. Since the hBDs were measured in duplicate, it would be useful to present some data showing the reproducibility of the measurements – e.g. CVs 

Answer: We do agree with this reviewer that this information is relevant and should be included in the text. We have added this information at the methods section in page 06. We correct the text including this information “Intra- and inter-assay coefficients of variation were <5% in this study. 

8. Were hBDs associated with any demographic/socio-behavioural factors assessed in this study? The authors should assess this and evaluate the relationship between hBDs and BV, adjusting for possible confounders and effect modifiers e.g. hormonal contraceptive use and smoking differ between comparison groups. 

Answer: We do agree with this reviewer that evaluate the relationship between hBDs and BV, adjusting for possible confounders. Please refer to the answer provided to question number 6. 

9. Page 9-11: The following conclusions should be reworded:

“Thus, the production of hBDs is related to the presence of Lactobacillus spp. in the vaginal microbiota.”

“but is mainly due to the lack of Lactobacillus species in the vaginal microbiota which could modulate the secretion of defensins”

“we suggest that the presence of Lactobacillus spp. in the vaginal microbiota plays an important role in the regulation of human beta defensins production in women of reproductive age.”

It is not possible, based on the data presented in this study (i.e. Nugent score) to conclude that hBD production is related to Lactobacillus spp. specifically. The relationship may have nothing to do with the lactobacilli, but may rather be due to downregulation or degradation of hBD production in the presence of non-optimal bacteria (which is discussed on page 11). It would also be useful to compare the score for the Lactobacillus morphotype to hBD concentrations – a sub-analysis could also be conducted to evaluate the relationship between Lactobacillus score and hBD concentrations among BV negative women only (i.e. when Gardnerella and Mobiluncus morphotypes are absent or rare, is hBD associated with Lactobacillus morphotype score?). 

Answer: We do agree with this reviewer that it is not possible, based on the data presented in this study (i.e. Nugent score) to conclude that hBD production is related to Lactobacillus spp. specifically. Then, as suggested, we performed the correlation between Nugent score and measured levels of hBDs1-3 were assessed statistically by calculating Spearman correlation coefficients (Spearman’s rho), for those participants from BV group. This analysis is presented in Results section page 10.

10. Page 10, line 217: Please clarify the following statement regarding “activation of the immune system” by Lactobacillus spp. since lactobacilli are also considered immunoregulatory. 

Answer:We thank this reviewer for pointing this issue. In fact, the text was not clear. We corrected the text for “and tolerance of the immune system” (page 12).

11. It would be preferable to refer to “optimal” microbiota, instead of “normal” microbiota. See McKinnon et al. AIDS Research and Human Retroviruses. 2019 Mar 1;35(3):219-28. 

Answer: We replaced “normal” microbiota for “optimal microbiota” throughout the manuscript.

Reviewer #2: The authors measured the levels of HBD1, 2 and 3 in CVL from 634 women by BV status defined by Nugent score. They show that the levels of all three defensins to be significantly lower in 317 women with BV (Nugent score of >7) than in 317 randomly selected women without BV (Nugent score 0-3). BV is associated with poor health outcomes and increased susceptibility to HIV which has been associated with pro-inflammatory responses and there is great interest in gaining a better understanding of the etiology.

It is argued that via their direct antimicrobial action, the beta defensins contribute to the protection of the vaginal microenvironment. There is circumstantial evidence that the innate immune response to resident Lactobacilli which includes anti-microbial peptides produced by the vaginal epithelium, contributes to protection against bacterial colonisation and subsequent dysbiosis associated with BV. The authors cite evidence that vaginal antimicrobial polypeptides can selectively allow for the growth of (resident) Lactobacilli, which is suggestive of a mechanism but the data is derived from a small number of women.

Existing data on the association with BV and levels of HBDs is limited and somewhat contradictory with some studies showing higher levels of HBD and some showing lower levels. HBDs recruit innate and adaptive immune cells and associated proinflammatory cytokines such as IL6 and IL8 have also been measured in some studies- again with conflicting results. In a review from 2014 which is not cited by the authors, Mitchell and Marazzo discuss the immune response associated with BV and any evidence for the modulating impact of Lactobacilli in the context of inflammatory markers (associated with increased susceptibility HIV). 

Answer:We agree with this reviewer that this review is important in this context and we included it in the discussion, page 11. 

They review data on the associations of the proinflammatory cytokines ILbeta and IL8 with BV and also discuss associations with antimicrobial peptides. There is no consistent finding and they conclude that the considerable variability in findings is probably due to the fact that most studies are small and underpowered and constrained by their cross-sectional design which does not account for the acknowledged (almost daily) fluctuations in vaginal microbiota (Gajer 2012). Longitudinal treatment studies are considered more robust and provide more consistent data with 5/6 showing reductions in levels of IL1beta and 3/6 reductions in IL8 after treatment for BV. Data on HBD was derived predominantly from pregnant women (Mitchell et al 2013) and showed that all 9 bacterial species associated with BV were associated with lower levels of HBD3 – but this was not true for HBD2. In a more direct approach, genetic analysis has also shed some light on factors associated with susceptibility to BV with polymorphisms in TLR IL6 and IL8 associated with increased susceptibility to BV arguing a role a robust innate immune response in protection. Whilst data on cytokines and BV has also been conflicting, this reviewer thinks that the paper would be strengthened by inclusion of data on the levels of proinflammatory cytokines such as IL1b, IL6 and IL8 (temos n=163 com IL1 e IL6 apenas para microbiota normal) as this would provide insights into the immunological consequences of the lower observed levels of HBD1, 2, 3 and potentially provide some mechanistic insights which have been suggested.

 Answer: We appreciate this point out by the reviewer and we agree that it will be an excellent opportunity for a new study, but, at this time, unfortunately we can`t assess this cytokines because most of the cervicovaginal samples were depleted during hBDs assays.

The etiology of BV is undoubtedly complex and multifactorial –without even considering the diversity in bacterial species which have been associated with both health and disease across different studies which complicates the interpretation of associations seen. The authors have not addressed this complexity and the gaps in their review of the literature reflects this (incluir diversidade da vb e que nossa forma de acessar a microbiota noa permite detectar essa heterogeneidade da VB). In their review Martin and Marazzo (JID 2014) discuss the importance of accounting for smoking, hormonal status and sexual behaviour in analyses (se basear nisso para fazer as subanalises da comparacao dos ,niveis de hbds entre os grupos) as these are all known to play a role (Mitchell and Murphy 2014). In particular, estrodiol has been shown to be reduce the production of HBD2 and SLP and mRNA expression of TLR by vaginal epithelial cells in vitro (Wira et al 2015) (citar isso na discussao). In their comprehensive paper of 2019, Fichorova et al also describe an important role for hormonal contraception in susceptibility to dysbiosis in the context of innate immune responses. Inflammatory responses are also known to fluctuate throughout the menstrual cycle with lower levels from day 7-10. 

Answer: All studies suggested by this reviewer were included in the discussion of the current version of the manuscript. We do appreciate all these enriching suggestions. In addition, we regret to inform that we cannot assess cytokines levels because samples were depleted. We do agree with these reviewer and consider this is an excellent opportunity for a new study.

 These authors show that levels of smoking and hormonal contraceptive use differ significantly by BV status and this could confound their analysis. The authors state that they cannot access any more information about hormonal contraceptive use, but they should at least discuss the limitations of their study in the context of available literature. At the very least they should run a multivariate analysis to accounts for smoking and hormonal status to see whether the associations with BV and HBD remain significant.

Answer: We do agree with this reviewer that evaluate the association between hBDs and BV, adjusting for possible confounders is relevant and should be included in the results. Thus, we performed a multivariable logistic regression analysis using a forward stepwise model selection process (variables retained at P-value ≤0.15) to test the variables independently associated with BV from sociodemographic and hBD1-3 data available. This analysis is presented in Results section pages 09-10.

It is acknowledged that BV is characterised by the general absence of “healthy Lactobacilli “ and it is debated whether the lactobacilli exert a positively protective effect. It has been shown for example that L crispatus and L jensenii are able to directly induce the production of HBD2 and thereby inhibit the growth of some gram negative bacteria. The reviewer understands that it is probably beyond the scope of the authors to undertake a comprehensive characterisation of the bacterial composition within their cohort, but this should be addressed within the discussion. 

Answer: We thank this reviewer for this interesting comment. We do agree that Nugent scoring does not allow the full characterization of the microbiota and should be acknowledged as a study limitation. Text added to the discussion section at pages 12-13)

In conclusion, this cross sectional study provides some interesting suggestive evidence that anti-microbial defensins might be associated with BV but limitations in the design mean that it provides limited additional extra value to the existing literature in its current form . The paper would be benefit from additional analysis to account for known confounders,a more thorough review of the literature discussing limitations, and if at all possible inclusion of additional data on levels of the pro-inflammatory cytokines which have been shown to be associated with BV.

Answer: We thank this reviewer for acknowledging our efforts. We do agree that the current studies has limitations (i. e. no cytokine data), but it should contribute with robust data that may contribute for better understanding the relation between BV and cervicovaginal hBDs.

6. PLOS authors have the option to publish the peer review history of their article (what does this mean?). If published, this will include your full peer review and any attached files.

Do you want your identity to be public for this peer review? For information about this choice, including consent withdrawal, please see our Privacy Policy.

Reviewer #1: No

Reviewer #2: No

---

## [Decision Letter · Decision Letter 1]

28 Oct 2021

PONE-D-21-03534R1Cervicovaginal levels of human beta defensins during bacterial vaginosisPLOS ONE

Dear Dr. Guimaraes da Silva,

Thank you for submitting your manuscript to PLOS ONE. After careful consideration, we feel that it has merit but does not fully meet PLOS ONE’s publication criteria as it currently stands. Therefore, we invite you to submit a revised version of the manuscript that addresses the points raised during the review process.

I feel that the manuscript's quality has been improved, but there are still some  inconsistencies to be fixed, as suggested by one of the referees Please submit your revised manuscript by Dec 12 2021 11:59PM. If you will need more time than this to complete your revisions, please reply to this message or contact the journal office at plosone@plos.org. Please include the following items when submitting your revised manuscript:A rebuttal letter that responds to each point raised by the academic editor and reviewer(s). You should upload this letter as a separate file labeled 'Response to Reviewers'.A marked-up copy of your manuscript that highlights changes made to the original version. You should upload this as a separate file labeled 'Revised Manuscript with Track Changes'.An unmarked version of your revised paper without tracked changes. You should upload this as a separate file labeled 'Manuscript'.If applicable, we recommend that you deposit your laboratory protocols in protocols.io to enhance the reproducibility of your results. Protocols.io assigns your protocol its own identifier (DOI) so that it can be cited independently in the future. For instructions see: https://journals.plos.org/plosone/s/submission-guidelines#loc-laboratory-protocols. Additionally, PLOS ONE offers an option for publishing peer-reviewed Lab Protocol articles, which describe protocols hosted on protocols.io. Read more information on sharing protocols at https://plos.org/protocols?utm_medium=editorial-email&utm_source=authorletters&utm_campaign=protocols.

We look forward to receiving your revised manuscript.

Kind regards,

Antonella Marangoni, Ph.D.

Academic Editor

PLOS ONE

Journal Requirements:

Reviewers' comments:

Reviewer's Responses to Questions

**Comments to the Author**

1. If the authors have adequately addressed your comments raised in a previous round of review and you feel that this manuscript is now acceptable for publication, you may indicate that here to bypass the “Comments to the Author” section, enter your conflict of interest statement in the “Confidential to Editor” section, and submit your "Accept" recommendation.

Reviewer #2: (No Response)

Reviewer #3: (No Response)

2. Is the manuscript technically sound, and do the data support the conclusions?

Reviewer #2: Partly

Reviewer #3: Yes

3. Has the statistical analysis been performed appropriately and rigorously? 

Reviewer #2: Yes

Reviewer #3: I Don't Know

4. Have the authors made all data underlying the findings in their manuscript fully available?

Reviewer #2: Yes

Reviewer #3: Yes

5. Is the manuscript presented in an intelligible fashion and written in standard English?

Reviewer #2: Yes

Reviewer #3: Yes

6. Review Comments to the Author

Reviewer #2: The manuscript is considerably improved and the authors have addressed many of the editorial comments. There are still some careless inconsistencies throughout and these should be addressed with thought for the overall flow of the discussion.

A major strength of the study lies in the number of individuals which have been included in the analysis, but the interpretation of the associations seen is hampered by the limited analysis performed so the authors should be careful not to over interpret their findings.

Specifically, the authors should caution against drawing firm conclusions about the association Lactobacilli spp and levels of hBD as they have not characterised the bacterial composition of the swabs. Statements such as

line 224 “ Thus, the production presence of hBDs is related to the presence of a increased abundance of Lactobacillus spp. in the vaginal microbiota “

should be modified accordingly to be more consistent with

line 283 “Thus, we postulate may hypothesize that the decreased lower levels of hBD-1, -2 and -3 in the presence of BV could be, in part, due to the production of proteases by the complex highly diverse bacterial community of BV,”

Importantly, a multivariate analysis has been carried out to explore the nature of the associations observed in the univariate analysis. The levels of HBD 1-3 remain significantly associated with BV as do the number of sexual partners and use of hormonal contraception. The authors discuss their findings in the context of previous data, and perhaps could suggest additional future analyses in their discussion which might provide mechanistic insights into the associations that they describe (line 289)

There remain some careless “typos”;

Line 172 The concentrations of hBD are still 1000X too low. These errors have been corrected elsewhere in the manuscript and this is a careless oversight.

Reviewer #3: The manuscript "Cervicovaginal levels of human beta defensins during bacterial vaginosis" addresses a complex and controversial issue that has been under research thoroughly for the last years: the relationship between vaginal microbiota and endogenous inflammatory factors and regulation.

I agreed with most of the comments and observations of the two previous reviewers and found that the authors have addressed them appropriately. After the corrections and additions made, the manuscript has significantly improved.

Even when the methodology used is simple and does not allow a more profound analysis (such as characterization of Lactobacillus species), the conclusions are valuable and contribute to this field's knowledge.

Thus, it's my recommendation to accept it for publication.

7. PLOS authors have the option to publish the peer review history of their article (what does this mean?). If published, this will include your full peer review and any attached files.

Reviewer #2: No

Reviewer #3: No

---

## [Author Response · Author response to Decision Letter 1]

3 Nov 2021

RESPONSE TO REVIEWERS

Journal Requirements:

Answer: We found a typo in the title of reference 45 which was corrected, and some formatting errors along the references which were corrected as well. We did not cite papers retracted. 

Comments to the Author

1. If the authors have adequately addressed your comments raised in a previous round of review and you feel that this manuscript is now acceptable for publication, you may indicate that here to bypass the “Comments to the Author” section, enter your conflict of interest statement in the “Confidential to Editor” section, and submit your "Accept" recommendation.

Reviewer #2: (No Response)

Reviewer #3: (No Response)

2. Is the manuscript technically sound, and do the data support the conclusions?

Reviewer #2: Partly

Reviewer #3: Yes

3. Has the statistical analysis been performed appropriately and rigorously?

Reviewer #2: Yes

Reviewer #3: I Don't Know

4. Have the authors made all data underlying the findings in their manuscript fully available?

Reviewer #2: Yes

Reviewer #3: Yes

5. Is the manuscript presented in an intelligible fashion and written in standard English?

Reviewer #2: Yes

Reviewer #3: Yes

6. Review Comments to the Author

Reviewer #2: The manuscript is considerably improved and the authors have addressed many of the editorial comments. There are still some careless inconsistencies throughout and these should be addressed with thought for the overall flow of the discussion.

A major strength of the study lies in the number of individuals which have been included in the analysis, but the interpretation of the associations seen is hampered by the limited analysis performed so the authors should be careful not to over interpret their findings.

Specifically, the authors should caution against drawing firm conclusions about the association Lactobacilli spp and levels of hBD as they have not characterised the bacterial composition of the swabs. Statements such as

line 224 “ Thus, the production presence of hBDs is related to the presence of a increased abundance of Lactobacillus spp. in the vaginal microbiota “

should be modified accordingly to be more consistent with

line 283 “Thus, we postulate may hypothesize that the decreased lower levels of hBD-1, -2 and -3 in the presence of BV could be, in part, due to the production of proteases by the complex highly diverse bacterial community of BV,”

Answer: We thank the reviewer for this pointing this out. Then, we removed this sentence “ Thus, the production presence of hBDs is related to the presence of a increased abundance of Lactobacillus spp. in the vaginal microbiota “.

Importantly, a multivariate analysis has been carried out to explore the nature of the associations observed in the univariate analysis. The levels of HBD 1-3 remain significantly associated with BV as do the number of sexual partners and use of hormonal contraception. The authors discuss their findings in the context of previous data, and perhaps could suggest additional future analyses in their discussion which might provide mechanistic insights into the associations that they describe (line 289)

Answer: We do agree with the reviewer that this information is relevant and should be included in the text. We just added this information at the discuss section in last paragraph.

There remain some careless “typos”;

Line 172 The concentrations of hBD are still 1000X too low. These errors have been corrected elsewhere in the manuscript and this is a careless oversight.

Answer: In fact we have detected some typos errors and have rewritten according. 

Reviewer #3: The manuscript "Cervicovaginal levels of human beta defensins during bacterial vaginosis" addresses a complex and controversial issue that has been under research thoroughly for the last years: the relationship between vaginal microbiota and endogenous inflammatory factors and regulation.

I agreed with most of the comments and observations of the two previous reviewers and found that the authors have addressed them appropriately. After the corrections and additions made, the manuscript has significantly improved.

Even when the methodology used is simple and does not allow a more profound analysis (such as characterization of Lactobacillus species), the conclusions are valuable and contribute to this field's knowledge.

Thus, it's my recommendation to accept it for publication.

7. PLOS authors have the option to publish the peer review history of their article (what does this mean?). If published, this will include your full peer review and any attached files.

Do you want your identity to be public for this peer review? For information about this choice, including consent withdrawal, please see our Privacy Policy.

Reviewer #2: No

Reviewer #3: No

---

## [Editor Report · Decision Letter 2]

17 Nov 2021

Cervicovaginal levels of human beta defensins during bacterial vaginosis

PONE-D-21-03534R2

Dear Dr. da Silva,

We’re pleased to inform you that your manuscript has been judged scientifically suitable for publication and will be formally accepted for publication once it meets all outstanding technical requirements.

Kind regards,

Antonella Marangoni, Ph.D.

Academic Editor

PLOS ONE
---

## [Editor Report · Acceptance letter]

22 Nov 2021

PONE-D-21-03534R2 

Cervicovaginal levels of human beta defensins during bacterial vaginosis 

Dear Dr. Silva:

I'm pleased to inform you that your manuscript has been deemed suitable for publication in PLOS ONE. Congratulations! Your manuscript is now with our production department. 

Kind regards, 

on behalf of

PhD Antonella Marangoni 

Academic Editor

PLOS ONE